# The Role of p.Ser1105Ser (in *NPHS1* Gene) and p.Arg548Leu (in *PLCE1* Gene) with Disease Status of Vietnamese Patients with Congenital Nephrotic Syndrome: Benign or Pathogenic?

**DOI:** 10.3390/medicina55040102

**Published:** 2019-04-12

**Authors:** Nguyen Thi Kim Lien, Pham Van Dem, Nguyen Thu Huong, Tran Minh Dien, Ta Thi Thu Thuy, Nguyen Van Tung, Nguyen Huy Hoang, Nguyen Thi Quynh Huong

**Affiliations:** 1Institute of Genome Research, Vietnam Academy of Science and Technology, 18, Hoang Quoc Viet str., Caugiay, Hanoi 100000, Vietnam; ntkimlienibt@gmail.com (N.T.K.L.); tungnv53@gmail.com (N.V.T.); 2University of Science, Vietnam National University, 334, Nguyen Trai str., Thanhxuan, Hanoi 100000, Vietnam; phamdemhd@gmail.com; 3Bachmai Hospital, Ministry of Health, Hanoi 100000, Vietnam; 4Vietnam National Children’s Hospital, Ministry of Health, 18/879 La Thanh str., Dongda, Hanoi 100000, Vietnam; Nguyenthuhuongnhp@gmail.com (N.T.H.); dientm@nhp.org.vn (T.M.D.); 5Hanoi Open University, Ministry of Education and Training, Hanoi 100000, Vietnam; thuthuycnsh@gmail.com; 6Faculty of Biotechnology, Graduate University of Science and Technology, Hanoi 100000, Vietnam; 7L’Hôpital Français de Hanoi, Ministry of Health, 1, Phuong Mai str., Dongda, Hanoi 100000, Vietnam; ntqhuong18@yahoo.com

**Keywords:** congential nephrotic syndrome (CNS), *NPHS1*, *NPHS2*, *PLCE1*, *WT1* mutation, Vietnamese patients

## Abstract

*Background and Objectives*: Congenital nephrotic syndrome (CNS), a genetic disease caused by mutations in genes on autosomes, usually occurs in the first three months after birth. A number of genetic mutations in genes, which encode for the components of the glomerular filtration barrier have been identified. We investigated mutations in *NPHS1*, *NPHS2*, *PLCE1 (NPHS3)*, and *WT1* genes that relate to the disease in Vietnamese patients. *Materials and Methods*: We performed genetic analysis of two unrelated patients, who were diagnosed with CNS in the Vietnam National Children’s Hospital with different disease status. The entire coding region and adjacent splice sites of these genes were amplified and sequenced using the Sanger method. The sequencing data were analyzed and compared with the *NPHS1, NPHS2, PLCE1,* and *WT1* gene sequences published in Ensembl (ENSG00000161270, ENSG00000116218, ENSG00000138193, and ENSG00000184937, respectively) using BioEdit software to detect mutations. *Results*: We detected a new variant p.Ser607Arg and two other (p.Glu117Lys and p.Ser1105Ser) in the *NPHS1* gene, as well as two variants (p.Arg548Leu, p.Pro1575Arg) in the *PLCE1* gene. No mutations were detected in the *NPHS2* and *WT1* genes. Patient 1, who presented a heterozygous genotype of p.Ser1105Ser and p.Arg548Leu had a mild disease status but patient 2, who presented a homozygous genotype of these alleles, had a severe phenotype. *Conclusions*: These results suggest that variants p.Ser1105Ser (in *NPHS1* gene) and p.Arg548Leu (in *PLCE1* gene) in the homozygous form might play a role in the development of the disease in patients.

## 1. Introduction

Congenital nephrotic syndrome (CNS) is a type of kidney disease that is characterized by elevated proteinuria and occurs in the first three months of life [1]. The clinical symptoms used as criteria for the diagnosis of CNS are: Edema, urinary protein >50 mg/24 h, and serum albumin <25 g/L. The cause of this syndrome are mutations in genes that encode for structural proteins of the kidney filter [2,3,4]. Mutations in the *NPHS1*, *NPHS2*, *PLCE1*, and *WT1* genes have been identified as the cause of the disease [5,6,7,8,9]. 

The gene *NPHS1* (OMIM 602716) is located on chromosome 19q13.1 [10,11] and consists of 29 exons, encoding for nephrin protein, a cell-surface protein of podocytes with 1241 amino acids [12]. Nephrin is a protein of the immunoglobulin (Ig) family. It plays an important role in the structure and the selective filtration function of the glomerular slit diaphragm [10,13]. Mutations in the *NPHS1* gene also lead to premature birth, with low birth weight and a placental weight over 25% of the newborn weight [6]. Mutations in the *NPHS1* gene were detected in 98% of Finnish cases and in 39–80% of non-Finnish cases [11,14,15,16,17]. More than 236 mutations have been described (http://www.hgmd.cf.ac.uk/ac/index.php), most of which lead to a severe clinical form of CNS [8]. 

The *NPHS2* gene (OMIM 604766) was identified in chromosome 1q25-31 and consists of 9 exons [18]. *NPHS2* encodes for podocin, an integral membrane protein in the podocyte foot process membrane of the kidney glomerulus [19,20]. It connects the plasma membrane (nephrin) and the cytoskeleton of the podocytes. Its protein molecular structure consists of three domains: A transmembrane domain forming a hairpin structure, a short extracellular domain (N-terminus) and a long cytoplasmic domain (C-terminus) [18]. Podocin may serve in the structural organization of the slit diaphragm (via interactions with nephrin at its C-terminus) and the regulation of its filtration function (via interactions with CD2AP as an adaptor) [19,21]. Mutations in *NPHS2* also affect the expression of nephrin, leading to CNS [12,18,22,23,24,25,26]. In a recent study conducted on 80 European families, *NPHS2* mutations were found in half of the CNS cases, and *NPHS1* mutations were detected in one-third of cases [14]. *NPHS2* mutations have also been found in patients with CNS across the globe [8]. To date, 171 mutations have been detected in the *NPHS2* gene locus (http://www.hgmd.cf.ac.uk/ac/index.php).

The *PLCE1* gene (OMIM 610725) is located on chromosome 10q23, consists of 31 exons and encodes for phospholipase C epsilon protein [27]. PLCE1 is an enzyme of the phospholipase family that catalyzes the hydrolysis of polyphosphoinositides, creating inositol-1,4,5-triphosphate (a secondary messenger), which is involved in cell growth and differentiation. Mutations in the *PLCE1* gene have been determined as the cause of CNS [28]. The pathogenetic role of *PLCE1* gene was confirmed in the *PLCE1* knockout zebrafish model [29]. There have been 38 mutations detected in the *PLCE1* gene (http://www.hgmd.cf.ac.uk/ac/index.php).

The *WT1* gene (OMIM 607102) causing Wilm’s tumor, in chromosome 11p13, consists of 10 exons. *WT1* mutations have been detected in some CNS patients with diffuse mesangial sclerosis. WT1 is a zinc finger transcription factor that participates in the development of renal organs [30]. In the fetal kidney, *WT1* expression is observed in renal vesicles and developing podocytes, and is lost in podocytes of collapsed glomeruli [31]. In particular, *WT1* mutations were found in patients with isolated kidney disease in the first three months of life [14,32]. *WT1* mutations have been described in most patients with Denys–Drash syndrome (DDS) [33]. This syndrome is characterized by the triad of DMS (diffuse mesangial sclerosis), male pseudohermaphroditism, and/or Wilm’s tumor. Isolated DMS (IDMS) is early-onset nephropathy and most often progresses to end-stage renal failure by three years of age [34]. IDMS is considered to be an important cause of CNS and nephrotic syndrome in infants. *WT1* heterozygous mutations have been reported by Jeanpierre et al. [32] in four of ten IDMS patients. A total of 143 mutations have been detected in the *WT1* gene (http://www.hgmd.cf.ac.uk/ac/index.php).

In this study, we identified mutations in the *NPHS1*, *NPHS2*, *PLCE1*, and *WT1* genes in two patients with CNS and members of their families by sequencing the entire coding region and adjacent splice sites of these genes. Information about mutations in the associated genes will contribute to a general understanding of the disease.

## 2. Materials and Methods

### 2.1. Study Subjects

Patient 1 is a 2.5-month-old girl, who was hospitalized with clinical features such as moderate edema, pneumonia, jaundice, infected with hepatitis B virus, and elevated hepatic enzyme. She is the first child in her family. She was a full-term normal delivery and the weight of the placenta was unknown. Her mother was infected with hepatitis B virus. The biochemical indices of the blood serum revealed 49 mg/dL protein (normal is 6–24 mg/dL), 4.5 mg/dL creatinine (normal is 0.5–1.2 mg/dL), and 12 g/L albumin (normal is 35–50 g/L). The results of electrolysis showed that blood sodium, potassium, and calcium were 129 mmol/L (normal is 135–145 mmol/L), 3.9 mmol/L (normal is 3.5–4.5 mmol/L), and 1.9 mmol/L (normal is 2.2–2.6 mmol/L), respectively. In addition, biochemical indices of the urine revealed 11.2 g/24 h protein (normal is 0–0.2 g/24 h). She was diagnosed with CNS in the Vietnam National Children’s Hospital. Her parents had a normal phenotype. 

Patient 2 is a 7-day-old boy, who was hospitalized with clinical features such as moderate edema, severe pneumonia. He is the second child in the family with an older brother who died of CNS at one-month-old. However, we did not collect a sample from his brother for genetic analysis. He was a full-term normal delivery and the weight of the placenta was unknown. The biochemical indices of the blood serum revealed 29 mg/dL protein (normal is 6–24 mg/dL), 3.5 mg/dL creatinine (normal is 0.5–1.2 mg/dL), and 6.8 g/L albumin (normal is 35–50 g/L). The results of electrolytes showed that blood sodium, potassium, and calcium were 132 mmol/L (normal is 135–145 mmol/L), 2.5 mmol/L (normal is 3.5–4.5 mmol/L), and 1.9 mmol/L (normal is 2.2–2.6 mmol/L), respectively. In addition, biochemical indices of the urine revealed 6.8 g/24 h protein (normal is 0–0.2 g/24 h). He was diagnosed with CNS in the Vietnam National Children’s Hospital. His parents had a normal phenotype. 

The written consent form were obtained from the patients.

### 2.2. Genetic Analysis

The genomic DNA was isolated from peripheral blood samples (including a sample from patients and their families) using a Qiagen DNA blood mini kit (QIAamp DNA Blood Mini preparation kits, Qiagen, Hilden, Germany) following the manufacturer’s guidelines. The DNA concentration was determined using a Thermo Scientific NanoDrop spectrophotometer (Waltham, MA, USA). All of the exons and exon-intron boundaries of the *NPHS1*, *NPHS2*, *PLCE1*, and *WT1* genes were amplified and analyzed by direct sequencing. The oligonucleotide primers were synthesized and purchased from IDT (Integrated DNA Technologies, Inc., Coralville, IA, USA) [11,28,32,35] for amplifying the *NPHS1*, *NPHS2*, *PLCE1* and *WT1* genes, respectively. Fifty nanograms of genomic DNA was subjected to 35 cycles of PCR amplification in a 25 μL volume consisting of 10X PCR buffer (Thermo), 10 µM concentration of each primer, 20 mM MgCl_2_, 10 μM dNTPs, and 5 U Dream Taq DNA polymerase (Thermo). DNA was denatured at 95 °C for 12 min followed by 35 cycles of denaturation for 1 min at 95 °C, annealing for 1 min at 53–65 °C, and extension for 1 min at 72 °C, with a final extension for 7 min at 72 °C. The PCR amplification was carried out on an Eppendorf Mastercycler EP gradient (USA Scientific, Inc., Ocala, FL, USA). 

The DNA sequencing was performed in both directions, initiated from forward and reverse primers, which had been used in an initial PCR reaction. The PCR products were purified with the Qiagen Purification kit (QIAquick PCR Purification Kit, Qiagen, Hilden, Germany) and sequenced on an ABI PRISM 3500 Genetic Analyser machine (Applied Biosystems Thermo Scientific, Foster, CA, USA). The Sequencing data were analyzed and compared with the *NPHS1*, *NPHS2*, *PLCE1* and *WT1* gene sequence published on Ensembl (ENSG00000161270, ENSG00000116218, ENSG00000138193, and ENSG00000184937, respectively) by using the BioEdit software version 7.0.9.0 to detect mutations.

### 2.3. Ethical Approval

All experiments performed in accordance with relevant guidelines and regulations based on the experimental protocol on human subjects which was approved by the Scientific Committee of Institute of Genome Research, Vietnam Academy of Science and Technology under reference number 15/QD-NCHG. 

## 3. Results

### 3.1. NPHS1 Analysis

We detected a new variant p.Ser607Arg, and two other (p.Glu117Lys and p.Ser1105Ser) in the *NPHS1* gene from two unrelated patients with CNS. The nucleotide change c.3315G>A (p.Ser1105Ser), at the fourth nucleotide on the 5’- end of exon 26, was found in both of the patients in our study. The variants p.Glu117Lys (c.349G>A, in exon 3) and p.Ser607Arg (c.1821C>A, in exon 14) in the *NPHS1* gene were found in patient 2. All of these variants were heterozygous, except p.Ser1105Ser in patient 2. These variants were inherited from their parents. Among them, variant p.Ser607Arg was identified as a novel variant and has not yet been published in [36,37].

### 3.2. PLCE1 Analysis

For the *PLCE1* gene, two variants (p.Arg548Leu and p.Pro1575Arg) were detected in the patients. The variant p.Arg548Leu (c.1643G>T) in exon 5 was identified in patient 1 (in heterozygous form) and patient 2 (in homozygous form). Patient 2 also carried another variant p.Pro1575Arg (c.4724C>G) in exon 19. 

### 3.3. NPHS2 and WT1 Analysis

No *NPHS2* and *WT1* mutations were detected in any of the four CNS patients in this study.

## 4. Discussion

### 4.1. Patient 1

Patient 1 was a mild case and the patient’s condition had been stable after the treatment time. Genetic analysis showed that she carried simultaneously, a heterozygous variant p.Ser1105Ser (c.3315G>A) in the cytosolic domain of NPHS1, a heterozygous variant p.Ser607Arg in NPHS1 (Figure 1) and another variant, p.Arg548Leu, in PLCE1 (Figure 2). A prediction of the pathogenicity of the variants was done by using PolyPhen 2 and Mutation Taster tools [38,39]. The results showed that this variant was benign, with a score of 0.26 (by PolyPhen 2 tool), and potentially pathogenic, with a score of 110 (by Mutation Taster tool). Research by Liu et al. [40] showed that mutations leading to amino acid changes may be the cause of folding errors of the protein molecule, which affected the intracellular transport function of the NPHS1 protein. In the study of Liu et al. [40], most of the CNS patients carried heterozygous mutations in the Ig extracellular region of the protein molecule. The results of this study indicated that mutations in the Ig functional regions have important effects on the function of the NPHS1 protein. However, there is no spatial structure of the nephrin protein, so it is not possible to predict the effect of this mutation on protein folding.

The common variant, p.Ser1105Ser (c.3315G>A), which does not result in an amino acid substitution, has previously been found in Finnish, Chinese and Japanese subjects [5,41,42,43]. Research by Machuca et al. [44] showed that milder cases, resulting from mutant *NPHS1*, either had two mutations in the cytoplasmic tail or two missense mutations in the extracellular domain, including at least one that preserved structure and function. These observations also suggested that mutations in the cytosolic region may have less effect on protein function. This result could explain the mild phenotype of the patient and the normal phenotype of her father (Table 1).

### 4.2. Patient 2

A heterozygous polymorphism p.Glu117Lys in exon 3, a heterozygous polymorphism p.Ser607Arg in exon 14, and a homozygous polymorphism p.Ser1105Ser (c.3315G>A) in exon 26 of the *NPHS1* gene were identified in patient 2 (Figure 3). p.Glu117Lys was reported by Lenkkeri et al. [11] as a single nucleotide polymorphism in a CNS cohort and has now been accepted as a well-known polymorphism. The patient also carried a homozygous variant c.3315G>A, located in the 4th position in exon 26 of the *NPHS1* gene, which can affect the pre-mRNA splicing process. This theory has been reinforced by the investigations of Wu and Hurst [45] who revealed that 5’-pathogenic single nucleotide polymorphisms (SNPs) might be more common and they may have a more drastic effect on the subsequent protein. 

Patient 2 also carried two of genetic changes in the *PLCE1* gene, including a homozygous variant p.Arg548Leu and a heterozygous variant p.Pro1575Arg (Figure 4). PLCE1 belongs to the Ras family genes, which encode membrane-associated molecular switches that bind GTP and GDP and slowly hydrolyze GTP to GDP [46]. A homozygous polymorphism p.Arg548Leu was located in a Ras-GEF domain, which may thus affect its capacity to activate the Ras protein. p.Arg548Leu and p.Arg1575Pro were also detected by Machuca et al. [44] in CNS patients and they were identified to play a role in the pathogenesis in these patients. In addition, Yilmaz et al. [47] reported a case with CNS, which had a homozygous variant p.Glu117Lys in NPHS1 as the only cause of the disease.

In our study, the combination of the two variants p.Ser1105Ser (in *NPHS1* gene) and p.Arg548Leu (in *PLCE1* gene), in the homozygous form, could be the cause of his severe symptoms. Thus, these variants in the heterozygous forms created only a mild phenotype in patient 1, and did not indicate the phenotype in their parents, but in the homozygous form had created a very severe phenotype in patient 2. These results suggested that a variant can become pathogenic when in a homozygous form. 

## 5. Conclusions

In our study, we detected three variants (p.Glu117Leu, p.Ser607Arg, and p.Ser1105Ser) in the *NPHS1* gene and two variants (p.Arg548Leu and p.Pro1575Arg) in the *PLCE1* gene. Among these, a variant (p.Ser607Arg in exon 14) was novel in the *NPHS1* gene. No mutations were detected in the *NPHS2* and *WT1* genes. Patient 1, with a heterozygous genotype of p.Ser1105Ser and p.Arg548Leu had a mild disease status but patient 2, with a homozygous genotype of these alleles, had a severe phenotype. The results in our study suggested that variants p.Ser1105Ser (in *NPHS1* gene) and p.Arg548Leu (in *PLCE1* gene) in the homozygous form might play a role in the development of the disease in the patients with CNS. 

## Figures and Tables

**Figure 1 medicina-55-00102-f001:**
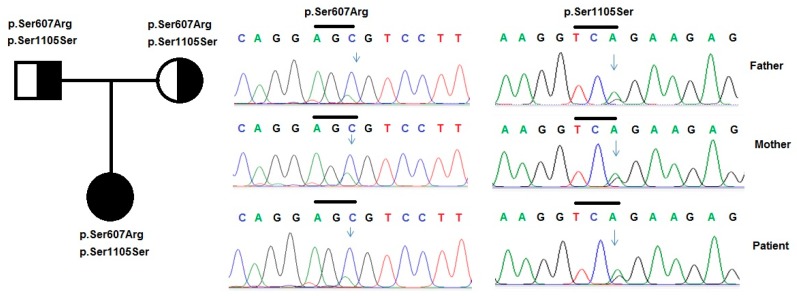
Variants in the *NPHS1* gene were identified by sequencing in patient 1 and her family. Pedigree of the patient’s family and two heterozygous variants p.Ser607Leu and p.Ser1105Ser in the *NPHS1* gene, that were identified in the patient and her father.

**Figure 2 medicina-55-00102-f002:**
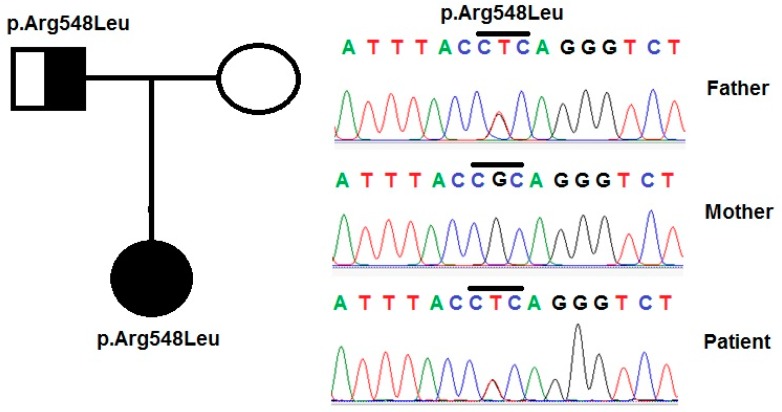
Variants in the *PLCE1* gene were identified by sequencing in patient 1 and her family. Pedigree of the patient’s family and variant p.Arg548Leu in the *PLCE1* gene, that was identified in the patient and her parents.

**Figure 3 medicina-55-00102-f003:**
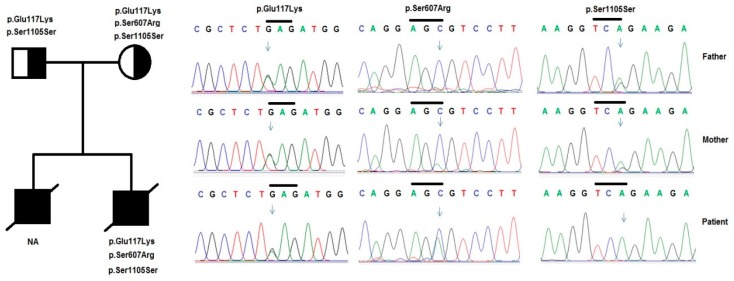
Variants in the *NPHS1* gene were identified by sequencing in patient 2 and his family. Pedigree of the patient’s family and variants in the *NPHS1* gene were identified, including a compound of heterozygous variants p.Glu117Lys, p.Ser607Arg, and a homozygous variant p.Ser1105Ser in the patient and his parents.

**Figure 4 medicina-55-00102-f004:**
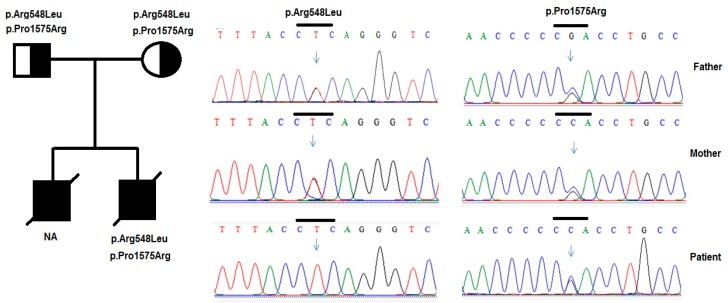
Variants in the *PLCE1* genes were identified by sequencing in patient 2 and his family. Pedigree of patient’s and two heterozygous variants p. Arg548Leu and p.Pro1575Arg in the *PLCE1* gene were identified in the patient and his parents.

**Table 1 medicina-55-00102-t001:** Summary the clinical and genetic data of patients with CNS in this study.

Patient	Sex/Age of Onset	Mutations in *NPHS1* Gene (exon)	Mutations in *PLCE1* Gene (exon)	Clinical Features	The Biochemical Indices of The Blood Serum and Urine			
					Protein	Creatinine	Albumin	Protein/24 h
Normal					6–24 mg/dL	0.5–1.2 mg/dL	35–50 g/L	0–0.2 g
Patient 1	Female/2.5-month-old	**p.Ser607Arg** (h) (exon 14) c.3315G>A (h) (exon 26)	p.Arg548Leu (h) (exon 5)	Moderate edema, pneumonia, jaundice, infected with hepatitis B virus	49 mg/dL	4.5 mg/dL	12 g/L	11.2 g
Patient 2	Male/7-day-old	p.Glu117Lys (h) (exon 3) **p.Ser607Arg** (h) (exon 14) c.3315G>A (H) (exon 26)	p.Arg548Leu (H) (exon 18) p.Pro1575Arg (h) (exon19)	Moderate edema, severe pneumonia	29 mg/dL	3.5 mg/dL	6.8 g/L	6.8 g

Bold letters are the novel mutations. h, heteozygous; H, homozygous.

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
