# Peer review of "The Role of p.Ser1105Ser (in NPHS1 Gene) and p.Arg548Leu (in PLCE1 Gene) with Disease Status of Vietnamese Patients with Congenital Nephrotic Syndrome: Benign or Pathogenic?"

_medicina, 2019, doi:10.3390/medicina55040102_

Round 1
Reviewer 1 Report
The authors present original and important data but the manuscript needs major grammatical corrections.

Author Response
Comments and Suggestions for Authors
The authors present original and important data but the manuscript needs major grammatical corrections.
Thank you for your comments. The manuscript have been edited in grammar as suggested by reviewer.

Reviewer 2 Report
In this report, the authors performed genotyping on two cases of congenital nephrotic syndrome and found a new variant (Ser607Arg), and two known mutations in NPHS1. In addition, they identified twwo variants in PLCE1 gene. In particular, patient 1 who has a milder phenotype is heterozygous, while patient 2 is homozygous for Ser1105Ser in NPHS1. They suggested that variants Ser1105Ser in NPHS1 and Arg548Leu in PLCE1 might play a role in the development of congenital nephrotic syndrome in these two paitents. While this is an interesting report, the conclusions however could not be sufficiently drawn from the results.
In patient 1, who has a milder phenotype, both parents have the same mutations (ser607Arg, Ser1105Ser) in NPHS1, yet neither of them has a phenotype. So these two mutations are unlikely responsible for the renal disease in patient 1.
Again in patient 1, both father and patient have Arg548Leu mutation in PLCE1, yet the father has no renal disease as described, so this mutation is not likely the cause of disease in patient.
In patient 2, who has a more server phenotype, all three members of the family have Glu117/Lys; both patient and mother have Ser607Arg. It is hard to believe that these two mutations are the culprit. For Ser1105Ser silent mutation, both parents are heterozygous, while the patient is homozygous. Though it is possible this NPHS1 homozygous mutation is involved in the development of renal phenotype, more evidence is necessary to confirm its causative role, moreover other genetic defects need to be excluded as well.
In the same token, both patient 2 and his parents have the same mutations in PLCE1, so these mutations are not enough to explain the phenotype.
Is it possible that there are other genetic mutations that account for the phenotype observed in these two patients.
There are typos throughout the report, so language needs to be improved.
Author Response
Comments and Suggestions for Authors
In this report, the authors performed genotyping on two cases of congenital nephrotic syndrome and found a new variant (Ser607Arg), and two known mutations in NPHS1. In addition, they identified two variants in PLCE1 gene. In particular, patient 1 who has a milder phenotype is heterozygous, while patient 2 is homozygous for Ser1105Ser in NPHS1. They suggested that variants Ser1105Ser in NPHS1 and Arg548Leu in PLCE1 might play a role in the development of congenital nephrotic syndrome in these two paitents. While this is an interesting report, the conclusions however could not be sufficiently drawn from the results.
In patient 1, who has a milder phenotype, both parents have the same mutations (Ser607Arg, Ser1105Ser) in NPHS1, yet neither of them has a phenotype. So these two mutations are unlikely responsible for the renal disease in patient 1.
Again in patient 1, both father and patient have Arg548Leu mutation in PLCE1, yet the father has no renal disease as described, so this mutation is not likely the cause of disease in patient.
Thank you for your comments. We agree with your comments. The genotype of the father and the patient is completely the same, but only the patient has the disease so we assume that the nucleotide change in the 4th position in exon 26 (p.Ser1105Ser) of the NPHS1 gene, and could affect the mRNA splicing process. This change may not have caused the disease phenotype in the patient’s parents but caused mild phenotype in the patient.
In patient 2, who has a more server phenotype, all three members of the family have Glu117/Lys; both patient and mother have Ser607Arg. It is hard to believe that these two mutations are the culprit. For Ser1105Ser silent mutation, both parents are heterozygous, while the patient is homozygous. Though it is possible this NPHS1 homozygous mutation is involved in the development of renal phenotype, more evidence is necessary to confirm its causative role, moreover other genetic defects need to be excluded as well.
In patient 1, with mild disease pattern, we detected a heterozygous change p.Ser1105Ser in NPHS1 gene and a heterozygous mutation p.Arg548Leu in PLCE1 gene. While in the patient 2, with severe phenotype, we identified a heterozygous polymorphism p.Gly117Lys, a homozygous change p.Ser1105Ser in NPHS1 gene and a homozygous mutation p.Arg548Leu in PLCE1 gene. So we assume about the pathogenic role of these mutations in the homozygous form in the patient.
In the same token, both patient 2 and his parents have the same mutations in PLCE1, so these mutations are not enough to explain the phenotype.
For the PLCE1 gene, the patient’s father carried a heterozygous mutation p.Arg548Leu while the patient had a homozygous mutation p.Arg548Leu.
Patient 2 also carried a homozygous change p.Ser1105Ser, which located in the 4th position in exon 26 of the NPHS1 gene, and could affect to the mRNA splicing process.
Therefore, in this paper we present an open issue to discuss about the role of a homozygous mutation p.Ser1105Ser in NPHS1 gene or a homozygous mutation p.Arg548Leu in PLCE1 gene or the combination of these mutations to make severe phenotype in the patient with CNS.
However, in order to confirm the pathogenic role of these mutations, more comprehensive studies are needed. In the context of this paper, we would like to give a small opinion that contributes to the general study of the disease.
Is it possible that there are other genetic mutations that account for the phenotype observed in these two patients.
We have excluded other possibilities that could be cause the disease such as mutations in NPHS2 and WT1 genes. All exons and exon – intron boundaries of these genes have been sequenced but no mutations have been identified.
There are typos throughout the report, so language needs to be improved.
The manuscript has been edited in grammar as suggested by reviewer.
